# Temporal Coherency based Criteria for Predicting Video Frames using Deep Multi-stage Generative Adversarial Networks

**Prateep Bhattacharjee[1], Sukhendu Das[2]**
Visualization and Perception Laboratory
Department of Computer Science and Engineering
Indian Institute of Technology Madras, Chennai, India
[1]`prateepb@cse.iitm.ac.in`, [2]`sdas@iitm.ac.in`

## Abstract

Predicting the future from a sequence of video frames has been recently a sought after yet challenging task in the field of computer vision and machine learning. Although there have been efforts for tracking using motion trajectories and flow features, the complex problem of generating unseen frames has not been studied extensively. In this paper, we deal with this problem using convolutional models within a multi-stage Generative Adversarial Networks (GAN) framework. The proposed method uses two stages of GANs to generate crisp and clear set of future frames. Although GANs have been used in the past for predicting the future, none of the works consider the relation between subsequent frames in the temporal dimension. Our main contribution lies in formulating two objective functions based on the Normalized Cross Correlation (NCC) and the Pairwise Contrastive Divergence (PCD) for solving this problem. This method, coupled with the traditional L1 loss, has been experimented with three real-world video datasets *viz.* Sports-1M, UCF-101 and the KITTI. Performance analysis reveals superior results over the recent state-of-the-art methods.

## 1  Introduction

Video frame prediction has recently been a popular problem in computer vision as it caters to a wide range of applications including self-driving cars, surveillance, robotics and in-painting. However, the challenge lies in the fact that, real-world scenes tend to be complex, and predicting the future events requires modeling of complicated internal representations of the ongoing events. Past approaches on video frame prediction include the use of recurrent neural architectures [19], Long Short Term Memory [8] networks [22] and action conditional deep networks [17]. Recently, the work of [14] modeled the frame prediction problem in the framework of Generative Adversarial Networks (GAN). Generative models, as introduced by Goodfellow *et. al.*, [5] try to generate images from random noise by simultaneously training a generator (G) and a discriminator network (D) in a process similar to a zero-sum game. Mathieu *et. al.* [14] shows the effectiveness of this adversarial training in the domain of frame prediction using a combination of two objective functions (along with the basic adversarial loss) employed on a multi-scale generator network. This idea stems from the fact that the original $L2$-loss tends to produce blurry frames. This was overcome by the use of Gradient Difference Loss (GDL) [14], which showed significant improvement over the past approaches when compared using similarity and sharpness measures. However, this approach, although producing satisfying results for the first few predicted frames, tends to generate blurry results for predictions far away ($\sim$6) in the future.

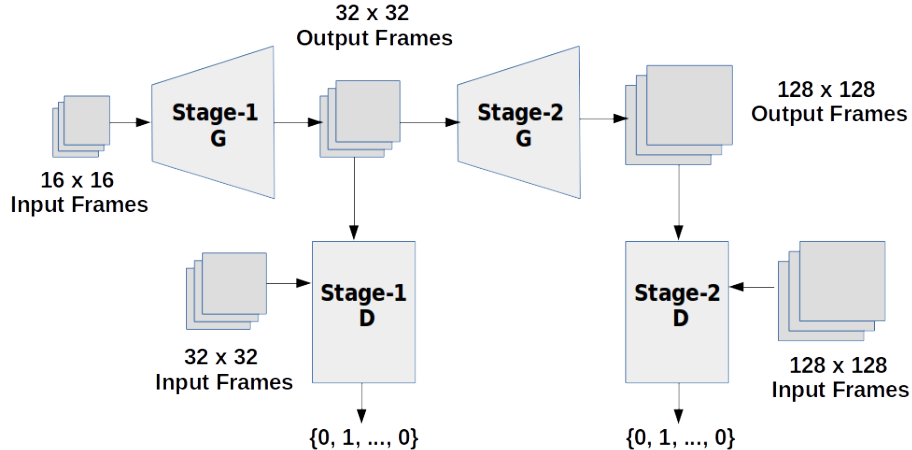

Figure 1: The proposed multi-stage GAN framework. The stage-1 generator network produces a low-resolution version of predicted frames which are then fed to the stage-2 generator. Discriminators at both the stages predict 0 or 1 for each predicted frame to denote its origin: synthetic or original.

In this paper, we aim to get over this hurdle of blurry predictions by considering an additional constraint between consecutive frames in the temporal dimension. We propose two objective functions: (a) **Normalized Cross-Correlation Loss (NCCL)** and (b) **Pairwise Contrastive Divergence Loss (PCDL)** for effectively capturing the inter-frame relationships in the GAN framework. NCCL maximizes the cross-correlation between neighborhood patches from consecutive frames, whereas, PCDL applies a penalty when subsequent generated frames are predicted wrongly by the discriminator network (D), thereby separating them far apart in the feature space. Performance analysis over three real world video datasets shows the effectiveness of the proposed loss functions in predicting future frames of a video.

The rest of the paper is organized as follows: section 2 describes the multi-stage generative adversarial architecture; sections 3 - 6 introduce the different loss functions employed: the adversarial loss (AL) and most importantly NCCL and PCDL. We show the results of our experiments on Sports-1M [10], UCF-101 [21] and KITTI [4] and compare them with state-of-the-art techniques in section 7. Finally, we conclude our paper highlighting the key points and future direction of research in section 8.

## 2 Multi-stage Generative Adversarial Model

Generative Adversarial Networks (GAN) [5] are composed of two networks: (a) the Generator (G) and (b) the Discriminator (D). The generator G tries to generate realistic images by learning to model the true data distribution $p_{data}$ and thereby trying to make the task of differentiating between original and generated images by the discriminator difficult. The discriminator D, in the other hand, is optimized to distinguish between the synthetic and the real images. In essence, this procedure of alternate learning is similar to the process of two player min-max games [5]. Overall, the GANs minimize the following objective function:

$$\min_G \max_D v(D,G) = \mathbb{E}_{x \sim p_{data}}[log(D(x))] + \mathbb{E}_{z \sim p_z}[log(1 - D(G(z)))] \qquad (1)$$

where, $x$ is a real image from the true distribution $p_{data}$ and $z$ is a vector sampled from the distribution $p_z$, usually to be uniform or Gaussian. The adversarial loss employed in this paper is a variant of that in equation 1, as the input to our network is a sequence of frames of a video, instead of a vector $z$.

As convolutions account only for short-range relationships, pooling layers are used to garner information from wider range. But, this process generates low resolution images. To overcome this, Mathieu *et. al.* [14] uses a multi-scale generator network, equivalent to the reconstruction process of a Laplacian pyramid [18], coupled with discriminator networks to produce high-quality output frames of size $32 \times 32$. There are two shortcomings of this approach:

a. Generating image output at higher dimensions *viz.* $(128 \times 128)$ or $(256 \times 256)$, requires multiple use of upsampling operations applied on the output of the generators. In our proposed model, this upsampling is handled by the generator networks itself implicitly through the use of consecutive unpooling operations, thereby generating predicted frames at much higher resolution in lesser number of scales.

b. As the generator network parameters are not learned with respect to any objective function which captures the temporal relationship effectively, the output becomes blurry after $\sim 4$ frames.

To overcome the first issue, we propose a multi-stage (2-stage) generative adversarial network (MS-GAN).

## 2.1 Stage-1

Generating the output frame(s) directly often produces blurry outcomes. Instead, we simplify the process by first generating crude, low-resolution version of the frame(s) to be predicted. The stage-1 generator $(G_1)$ consists of a series of convolutional layers coupled with unpooling layers [25] which upsample the frames. We used ReLU non-linearity in all but the last layer, in which case, hyperbolic tangent (tanh) was used following the scheme of [18]. The inputs to $G_1$ are $m$ number of consecutive frames of dimension $W_0 \times H_0$, whereas the outputs are $n$ predicted frames of size $W_1 \times H_1$, where, $W_1 = W_0 \times 2$ and $H_1 = H_0 \times 2$. These outputs, stacked with the upsampled version of the original input frames, produce the input of dimension $(m + n) \times W_1 \times H_1$ for the stage-1 discriminator $(D_1)$. $D_1$ applies a chain of convolutional layers followed by multiple fully-connected layers to finally produce an output vector of dimension $(m + n)$, consisting of 0's and 1's.

One of the key differences of our proposed GAN framework with the conventional one [5]is that, the discriminator network produces decision output for multiple frames, instead of a single $0/1$ outcome. This is exploited by one of the proposed objective functions, the PCDL, which is described later in section 4.

## 2.2 Stage-2

The second stage network closely resembles the stage-1 architecture, differing only in the input and output dimensions. The input to the stage-2 generator $(G_2)$ is formed by stacking the predicted frames and the upsampled inputs of $G_1$, thereby having dimension of $(m + n) \times W_1 \times H_1$. The output of $G_2$ are $n$ predicted high-resolution frames of size $W_2 \times H_2$, where, $W_2 = W_1 \times 4$ and $H_2 = H_1 \times 4$. The stage-2 discriminator $(D_2)$, works in a similar fashion as $D_1$, producing an output vector of length $(m + n)$.

Effectively, the multi-stage model can be represented by the following recursive equations:

$$\hat{Y}_k = \begin{cases} G_k(\hat{Y}_{k-1}, X_{k-1}), & for \quad k \geq 2 \\ G_k(X_{k-1}) & for \quad k = 1 \end{cases} \tag{2}$$

where, $\hat{Y}_k$ is the set of predicted frames and $X_k$ are the input frames at the $k$th stage of the generator network $G_k$.

## 2.3 Training the multi-stage GAN

The training procedure of the multi-stage GAN model follows that of the original generative adversarial networks with minor variations. The training of the discriminator and the generator are described as follows:

**Training of the discriminator** Considering the input to the discriminator $(D)$ as $X$ (series of $m$ frames) and the target output to be $Y$ (series of $n$ frames), $D$ is trained to distinguish between synthetic and original inputs by classifying $(X, Y)$ into class 1 and $(X, G(X))$ into class 0. Hence, for each of the $k$ stages, we train $D$ with target $\vec{1}$ (Vector of 1's with dimension $m$) for $(X, Y)$ and

target $\vec{0}$ (Vector of 0's with dimension $n$) for $(X, G(X))$. The loss function for training $D$ is:

$$\mathcal{L}_{adv}^{D} = \sum_{k=1}^{N_{stages}} L_{bce}(D_k(X_k, Y_k), \vec{1}) + L_{bce}(D_k(X_k, G_k(X_k)), \vec{0}) \qquad (3)$$

where, $L_{bce}$, the binary cross-entropy loss is defined as:

$$L_{bce}(A, A') = -\sum_{i=1}^{|A|} A'^i log(A^i) + (1 - A'^i)log(1 - A^i), A^i \in \{0, 1\}, A'^i \in [0, 1] \qquad (4)$$

where, $A$ and $A'$ are the target and discriminator outputs respectively.

**Training of the generator**    We perform an optimization step on the generator network $(G)$, keeping the weights of $D$ fixed, by feeding a set of consecutive frames $X$ sampled from the training data with target $Y$ (set of ground-truth output frames) and minimize the following adversarial loss:

$$\mathcal{L}_{adv}^{G}(X) = \sum_{k=1}^{N_{stages}} L_{bce}(D_k(X_k, G_k(X_k)), \vec{1}) \qquad (5)$$

By minimizing the above two loss criteria (eqns. 3, 5), $G$ makes the discriminator believe that, the source of the generated frames is the input data space itself. Although this process of alternate optimization of $D$ and $G$ is reasonably well designed formulation, in practical purposes, this produces an unstable system where $G$ generates samples that consecutively move far away from the original input space and in consequence $D$ distinguishes them easily. To overcome this instability inherent in the GAN principle and the issue of producing blurry frames defined in section 2, we formulate a pair of objective criteria: (a) Normalized Cross Correlation Loss (NCCL) and (b)Pairwise Contrastive Divergence Loss (PCDL), to be used along with the established adversarial loss (refer eqns. 3 and 5).

## 3    Normalized Cross-Correlation Loss (NCCL)

The main advantage of video over image data is the fact that, it offers a far richer space of data distribution by adding the temporal dimension along with the spatial one. Convolutional Neural Networks (CNN) can only capture short-range relationships, a small part of the vast available information, from the input video data, that too in the spatial domain. Although this can be somewhat alleviated by the use of 3D convolutions [9], that increases the number of learn-able parameters immensely. Normalized cross-correlation has been used since long time in the field of video analytics [1, 2, 16, 13, 23] to model the spatial and temporal relationships present in the data.

Normalized cross correlation (NCC) measures the similarity of two image patches as a function of the displacement of one relative to the other. This can be mathematically defined as:

$$NCC(f, g) = \sum_{x,y} \frac{(f(x, y) - \mu_f)(g(x, y) - \mu_g)}{\sigma_f \sigma_g} \qquad (6)$$

where, $f(x, y)$ is a sub-image, $g(x, y)$ is the template to be matched, $\mu_f, \mu_g$ denotes the mean of the sub-image and the template respectively and $\sigma_f, \sigma_g$ denotes the standard deviation of $f$ and $g$ respectively.

In the domain of video frame(s) prediction, we incorporate the NCC by first extracting small non-overlapping square patches of size $h \times h$ ($1 < h \leq 4$), denoted by a 3-tuple $P_t\{x, y, h\}$, where, $x$ and $y$ are the co-ordinates of the top-left pixel of a particular patch, from the predicted frame at time $t$ and then calculating the cross-correlation score with the patch extracted from the ground truth frame at time $(t - 1)$, represented by $\hat{P}_{t-1}\{x - 2, y - 2, h + 4\}$.

In simpler terms, we estimate the cross-correlation score between a small portion of the current predicted frame and the local neighborhood of that in the previous ground-truth frame. We assume that, the motion features present in the entire scene (frame) be effectively approximated by adjacent spatial blocks of lower resolution,using small local neighborhoods in the temporal dimension. This stems from the fact that, unless the video contains significant jitter or unexpected random events like

---

**Algorithm 1:** Normalized cross-correlation score for estimating similarity between a set of predicted frame(s) and a set of ground-truth frame(s).

---

**Input**: Ground-truth frames ($GT$), Predicted frames ($PRED$)
**Output**: Cross-correlation score ($Score_{NCC}$)
// $h =$ height and width of an image patch
// $H =$ height and width of the predicted frames
// $t =$ current time
// $T =$ Number of frames predicted
**Initialize:** $Score_{NCC} = 0$;
**for** $t = 1$ *to* $T$ **do**
   **for** $i = 0$ *to* $H$, $i \leftarrow i + h$ **do**
      **for** $j = 0$ *to* $H$, $j \leftarrow j + h$ **do**
         $P_t \leftarrow extract\_patch(PRED_t, i, j, h)$;
         /* Extracts a patch from the predicted frame at time $t$ of dimension $h \times h$ starting from the top-left pixel index $(i, j)$ */
         $\hat{P}_{t-1} \leftarrow extract\_patch(GT_{t-1}, i - 2, j - 2, h + 4)$;
         /* Extracts a patch from the ground-truth frame at time $(t-1)$ of dimension $(h+4) \times (h+4)$ starting from the top-left pixel index $(i - 2, j - 2)$ */
         $\mu_{P_t} \leftarrow avg(P_t)$;
         $\mu_{\hat{P}_{t-1}} \leftarrow avg(\hat{P}_{t-1})$;
         $\sigma_{P_t} \leftarrow standard\_deviation(P_t)$;
         $\sigma_{\hat{P}_{t-1}} \leftarrow standard\_deviation(\hat{P}_{t-1})$;
         $Score_{NCC} \leftarrow Score_{NCC} + max\left(0, \sum_{x,y} \frac{(P_t(x,y) - \mu_{P_t})(\hat{P}_{t-1}(x,y) - \mu_{\hat{P}_{t-1}})}{\sigma_{P_t}\sigma_{\hat{P}_{t-1}}}\right)$;
      **end**
   **end**
   $Score_{NCC} \leftarrow Score_{NCC}/\lfloor H/h \rfloor^2$;         // Average over all the patches
**end**
$Score_{NCC} \leftarrow Score_{NCC}/(T-1)$;         // Average over all the frames

---

scene change, the motion features remain smooth over time. The step-by-step process for finding the cross-correlation score by matching local patches of predicted and ground truth frames is described in algorithm 1.

The idea of calculating the NCC score is modeled into an objective function for the generator network $G$, where it tries to maximize the score over a batch of inputs. In essence, this objective function models the temporal data distribution by smoothing the local motion features generated by the convolutional model. This loss function, $\mathcal{L}_{NCCL}$, is defined as:

$$\mathcal{L}_{NCCL}(Y, \hat{Y}) = -Score_{NCC}(Y, \hat{Y}) \tag{7}$$

where, $Y$ and $\hat{Y}$ are the ground truth and predicted frames and $Score_{NCC}$ is the average normalized cross-correlation score over all the frames, obtained using the method as described in algorithm 1. The generator tries to minimize $\mathcal{L}_{NCCL}$ along with the adversarial loss defined in section 2.

We also propose a variant of this objective function, termed as Smoothed Normalized Cross-Correlation Loss (SNCCL), where the patch similarity finding logic of NCCL is extended by convolving with Gaussian filters to suppress transient (sudden) motion patterns. A detailed discussion of this algorithm is given in sec. A of the supplementary document.

## 4 Pairwise Contrastive Divergence Loss (PCDL)

As discussed in sec. 3, the proposed method captures motion features that vary slowly over time. The NCCL criteria aims to achieve this using local similarity measures. To complement this in a global scale, we use the idea of pairwise contrastive divergence over the input frames. The idea of exploiting this temporal coherence for learning motion features has been studied in the recent past [6, 7, 15].

By assuming that, motion features vary slowly over time, we describe $\hat{Y}_t$ and $\hat{Y}_{t+1}$ as a temporal pair, where, $\hat{Y}_i$ and $\hat{Y}_{t+1}$ are the predicted frames at time $t$ and $(t+1)$ respectively, if the outputs of the discriminator network $D$ for both these frames are 1. With this notation, we model the slowness principle of the motion features using an objective function as:

$$
\begin{aligned}
\mathcal{L}_{PCDL}(\hat{Y}, \vec{p}) &= \sum_{i=0}^{T-1} D_\delta(\hat{Y}_i, \hat{Y}_{i+1}, p_i \times p_{i+1}) \\
&= \sum_{i=0}^{T-1} p_i \times p_{i+1} \times d(\hat{Y}_i, \hat{Y}_{i+1}) + (1 - p_i \times p_{i+1}) \times max(0, \delta - d(\hat{Y}_i, \hat{Y}_{i+1}))
\end{aligned}
\tag{8}
$$

where, $T$ is the time-duration of the frames predicted, $p_i$ is the output decision ($p_i \in \{0, 1\}$) of the discriminator, $d(x, y)$ is a distance measure ($L2$ in this paper) and $\delta$ is a positive margin. Equation 8 in simpler terms, minimizes the distance between frames that have been predicted correctly and encourages the distance in the negative case, up-to a margin $\delta$.

## 5 Higher Order Pairwise Contrastive Divergence Loss

The Pairwise Contrastive Divergence Loss (PCDL) discussed in the previous section takes into account (dis)similarities between two consecutive frames to bring them further (or closer) in the spatio-temporal feature space. This idea can be extended for higher order situations involving three or more consecutive frames. For $n = 3$, where $n$ is the number of consecutive frames considered, PCDL can be defined as:

$$
\begin{aligned}
\mathcal{L}_{3-PCDL} &= \sum_{i=0}^{T-2} D_\delta(|\hat{Y}_i - \hat{Y}_{i+1}|, |\hat{Y}_{i+1} - \hat{Y}_{i+2}|, p_{i,i+1,i+2}) \\
&= \sum_{i=0}^{T-2} p_{i,i+1,i+2} \times d(|\hat{Y}_i - \hat{Y}_{i+1}|, |\hat{Y}_{i+1} - \hat{Y}_{i+2}|) \\
&\quad + (1 - p_{i,i+1,i+2}) \times max(0, \delta - d(|(\hat{Y}_i - \hat{Y}_{i+1})|, |(\hat{Y}_{i+1} - \hat{Y}_{i+2})|))
\end{aligned}
\tag{9}
$$

where, $p_{i,i+1,i+2} = 1$ only if $p_i, p_{i+1}$ and $p_{i+2}$- all are simultaneously 1, *i.e.*, the discriminator is very sure about the predicted frames, that they are from the original data distribution. All the other symbols bear standard representations defined in the paper.

This version of the objective function, in essence shrinks the distance between the predicted frames occurring sequentially in a temporal neighborhood, thereby increasing their similarity and maintaining the temporal coherency.

## 6 Combined Loss

Finally, we combine the objective functions given in eqns. 5 - 8 along with the general $L1$-loss with different weights as:

$$
\begin{aligned}
\mathcal{L}_{Combined} =& \lambda_{adv}\mathcal{L}_{adv}^G(X) + \lambda_{L1}\mathcal{L}_{L1}(X, Y) + \lambda_{NCCL}\mathcal{L}_{NCCL}(Y, \hat{Y}) \\
&+ \lambda_{PCDL}\mathcal{L}_{PCDL}(\hat{Y}, \vec{p}) + \lambda_{PCDL}\mathcal{L}_{3-PCDL}(\hat{Y}, \vec{p})
\end{aligned}
\tag{10}
$$

All the weights *viz.* $\lambda_{L1}, \lambda_{NCCL}, \lambda_{PCDL}$ and $\lambda_{3-PCDL}$ have been set as $0.25$, while $\lambda_{adv}$ equals $0.01$. This overall loss is minimized during the training stage of the multi-stage GAN using Adam optimizer [11].

We also evaluate our models by incorporating another loss function described in section A of the supplementary document, the Smoothed Normalized Cross-Correlation Loss (SNCCL). The weight for SNCCL, $\lambda_{SNCCL}$ equals $0.33$ while $\lambda_{3-PCDL}$ and $\lambda_{PCDL}$ is kept at $0.16$.

## 7 Experiments

Performance analysis with experiments of our proposed prediction model for video frame(s) have been done on video clips from Sports-1M [10], UCF-101 [21] and KITTI [4] datasets. The input-output configuration used for training the system is as follows: **input**: 4 frames and **output**: 4 frames.

We compare our results with recent state-of-the-art methods using two popular metrics: (a) **Peak Signal to Noise Ratio (PSNR)** and (b) **Structural Similarity Index Measure (SSIM)** [24].

## 7.1 Datasets

**Sports-1M**    A large collection of sports videos collected from YouTube spread over 487 classes. The main reason for choosing this dataset is the amount of movement in the frames. Being a collection of sports videos, this has sufficient amount of motion present in most of the frames, making it an efficient dataset for training the prediction model. Only this dataset has been used for training all throughout our experimental studies.

**UCF-101**    This dataset contains 13320 annotated videos belonging to 101 classes having 180 frames/video on average. The frames in this video do not contain as much movement as the Sports-1m and hence this is used only for testing purpose.

**KITTI**    This consists of high-resolution video data from different road conditions. We have taken raw data from two categories: (a) city and (b) road.

## 7.2 Architecture of the network

Table 1: Network architecture details; $G$ and $D$ represents the generator and discriminator networks respectively. $U$ denotes an unpooling operation which upsamples an input by a factor of 2.

| Network | Stage-1 (G) | Stage-2 (G) | Stage-1 (D) | Stage-2 (D) |
|---------|-------------|-------------|-------------|-------------|
| Number of feature maps | 64, 128, 256U, 128, 64 | 64, 128U, 256, 512U, 256, 128, 64 | 64, 128, 256 | 128, 256, 512, 256, 128 |
| Kernel sizes | 5, 3, 3, 3, 5 | 5, 5, 5, 5, 5, 5, 5 | 3, 5, 5 | 7, 5, 5, 5, 5 |
| Fully connected | N/A | N/A | 1024, 512 | 1024, 512 |

The architecture details for the generator (G) and discriminator (D) networks used for experimental studies are shown in table 1. All the convolutional layers except the terminal one in both stages of $G$ are followed by ReLU non-linearity. The last layer is tied with tanh activation function. In both the stages of $G$, we use unpooling layers to upsample the image into higher resolution in magnitude of 2 in both dimensions (height and width). The learning rate is set to $0.003$ for $G$, which is gradually decreased to $0.0004$ over time. The discriminator (D) uses ReLU non-linearities and is trained with a learning rate of $0.03$. We use mini-batches of 8 clips for training the overall network.

## 7.3 Evaluation metric for prediction

Assessment of the quality of the predicted frames is done by two methods: (a) Peak Signal to Noise Ratio (PSNR) and (b) Structural Similarity Index Measure (SSIM). **PSNR** measures the quality of the reconstruction process through the calculation of Mean-squared error between the original and the reconstructed signal in logarithmic decibel scale [1]. **SSIM** is also an image similarity measure where, one of the images being compared is assumed to be of perfect quality [24].

As the frames in videos are composed of foreground and background, and in most cases the background is static (not the case in the KITTI dataset, as it has videos taken from camera mounted on a moving car), we extract random sequences of $32 \times 32$ patches from the frames with significant motion. Calculation of motion is done using the optical flow method of *Brox et. al.* [3].

## 7.4 Comparison

We compare the results on videos from UCF-101, using the model trained on the Sports-1M dataset. Table 2 demonstrates the superiority of our method over the most recent work [14]. We followed similar choice of test set videos as in [14] to make a fair comparison. One of the impressive facts in our model is that, it can produce acceptably good predictions even in the 4th frame, which is a significant result considering that [14] uses separate smaller multi-scale models for achieving this

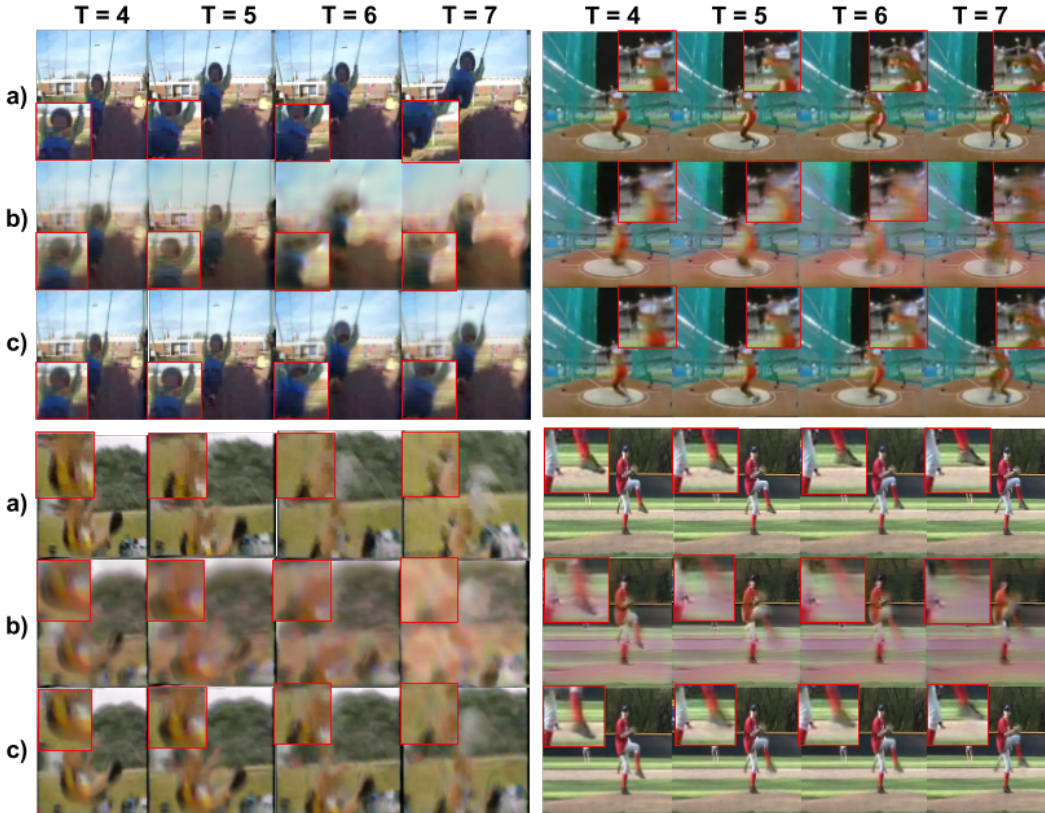

Figure 2: Qualitative results of using the proposed framework for predicting frames in UCF-101 with the three rows representing (a) Ground-truth, (b) Adv + L1 and (c) Combined (section 6) respectively. 'T' denotes the time-step. Figures in insets show zoomed-in patches for better visibility of areas involving motion (Best viewed in color).

feat. Also note that, even though the metrics for the first predicted frame do not differ by a large margin compared to the results from [14] for higher frames, the values decrease much slowly for the models trained with the proposed objective functions (rows 8-10 of table 2). The main reason for this phenomenon in our proposed method is the incorporation of the temporal relations in the objective functions, rather than learning only in the spatial domain.

Similar trend was also found in case of the KITTI dataset. We could not find any prior work in the literature reporting findings on the KITTI dataset and hence compared only with several of our proposed models. In all the cases, the performance gain with the inclusion of NCCL and PCDL is evident.

Finally, we show the prediction results obtained on both the UCF-101 and KITTI in figures 2 and 3. It is evident from the sub-figures that, our proposed objective functions produce impressive quality frames while the models trained with L1 loss tends to output blurry reconstruction. The supplementary document contains visual results (shown in figures C.1-C.2) obtained in case of predicting frames far-away from the current time-step (8 frames).

## 8   Conclusion

In this paper, we modified the Generative Adversarial Networks (GAN) framework with the use of unpooling operations and introduced two objective functions based on the normalized cross-correlation (NCCL) and the contrastive divergence estimate (PCDL), to design an efficient algorithm for video frame(s) prediction. Studies show significant improvement of the proposed methods over the recent published works. Our proposed objective functions can be used with more complex networks involving 3D convolutions and recurrent neural networks. In the future, we aim to learn weights for the cross-correlation such that it focuses adaptively on areas involving varying amount of motion.

Table 2: Comparison of performance for different methods using PSNR/SSIM scores for the UCF-101 and KITTI datasets. The first five rows report the results from [14]. (*) indicates models fine tuned on patches of size $64 \times 64$ [14]. (-) denotes unavailability of data. GDL stands for Gradient Difference Loss [14]. SNCCL is discussed in section A of the supplementary document. Best results in bold.

| Methods | 1st frame prediction score | | 2nd frame prediction score | | 4th frame prediction score | |
| --- | --- | --- | --- | --- | --- | --- |
| | UCF | KITTI | UCF | KITTI | UCF | KITTI |
| L1 | 28.7/0.88 | - | 23.8/0.83 | - | - | - |
| GDL L1 | 29.4/0.90 | - | 24.9/0.84 | - | - | - |
| GDL L1* | 29.9/0.90 | - | 26.4/0.87 | - | - | - |
| Adv + GDL fine-tuned* | 32.0/0.92 | - | 28.9/0.89 | - | - | - |
| Optical flow | 31.6/0.93 | - | 28.2/0.90 | - | - | - |
| Next-flow [20] | 31.9/- | - | - | - | - | - |
| Deep Voxel Flow [12] | 35.8/0.96 | - | - | - | - | - |
| Adv + NCCL + L1 | 35.4/0.94 | 37.1/0.91 | 33.9/0.92 | 35.4/0.90 | 28.7/0.75 | 27.8/0.75 |
| Combined | 37.3/**0.95** | 39.7/0.93 | 35.7/0.92 | 37.1/**0.91** | 30.2/**0.76** | 29.6/0.76 |
| **Combined + SNCCL** | **38.2/0.95** | **40.2/0.94** | **36.8/0.93** | **37.7/0.91** | **30.9/0.77** | **30.4/0.77** |
| Combined + SNCCL (full frame) | 37.3/0.94 | 39.4/0.94 | 35.1/0.91 | 36.4/0.91 | 29.5/0.75 | 29.1/0.76 |

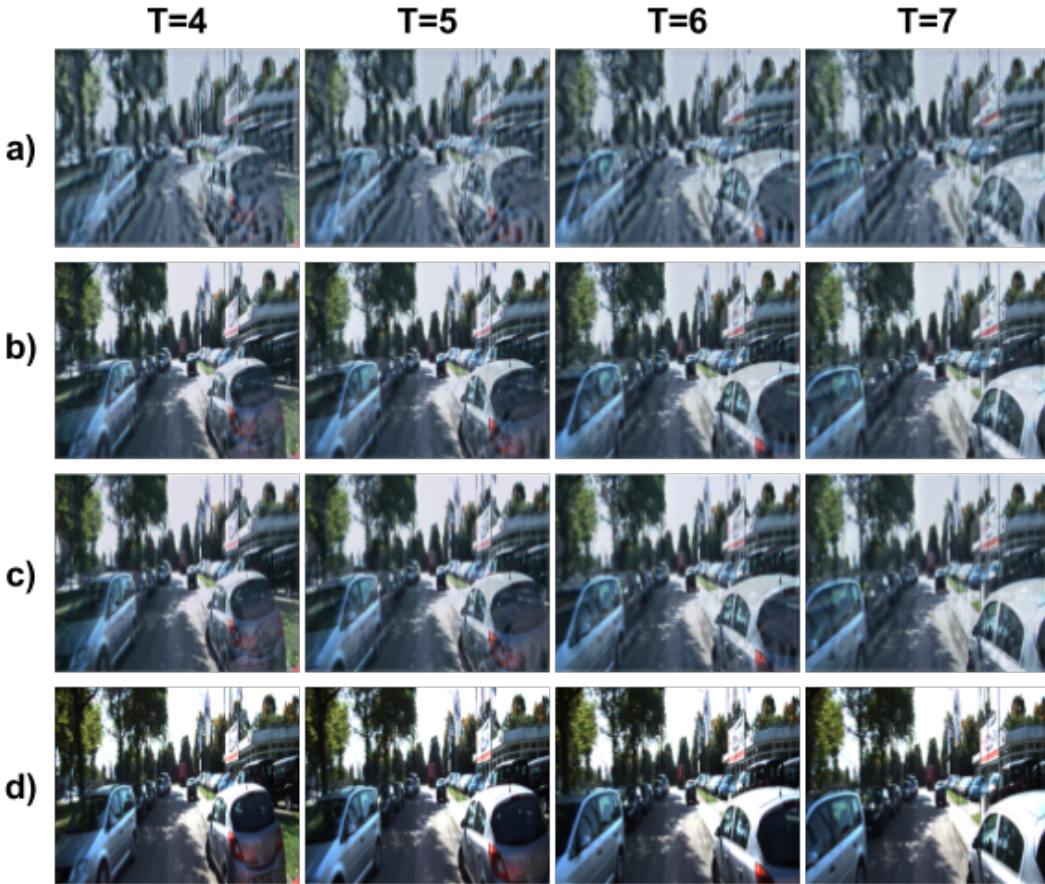

Figure 3: Qualitative results of using the proposed framework for predicting frames in the KITTI Dataset, for (a) L1, (b) NCCL (section 3), (c) Combined (section 6) and (d) ground-truth (Best viewed in color).

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
