[Supplementary Material · supplementary_note.pdf]

# Supplementary: Temporal Coherency based Criteria for Predicting Video Frames using Deep Multi-stage Generative Adversarial Networks

## A Smoothed Normalized Cross-Correlation Loss (SNCCL)

In this section, we provide a modification of the Normalized Cross-Correlation Loss (NCCL) presented in section 3 of the paper. This modification assumes that, while comparing two local patches between the previous frame at timestamp $t-1$ and the current frame at timestamp $t$, majority of the motion similar to both the frames occur around the central pixel of the patches. This assumption makes the system more *robust* to sudden small variation in motion occurring at the boundaries of the local patches.

To accomplish this heuristic in practical terms, a weight function can be learned whose parameters are learned adaptively. This requires learning these parameters along with those of the multi-stage GAN discussed in sec. 2, which is a non-trivial problem. For the sake of simplicity, we approximate this weight function using a two-dimensional mean-centered Gaussian low-pass filter (2D-GLPF) and experiment with varying amount of standard deviation of the filter. The algorithm for calculating the smoothed normalized cross-correlation score is outlined in algo. A.1.

This version of the algorithm automatically filters out areas containing zero or insignificant motion by calculating the L2 distance between two image patches (step $8$ of algo. A.1). This simple trick stops the value of the normalized cross-correlation score from shooting up due to high similarity in static areas, thereby letting the algorithm focus on predicting the actual motion patterns, rather than learning to produce static background (which is done by copying the pixel values). From line $8$ of algo A.1:

$$\Psi = max\big(0, d(\hat{P}_{t-1}, P_t^{'}) - \xi\big) \tag{I}$$

where, $\Psi$ is the indicator variable for finding a highly similar image patch, $\xi$ is the margin of similarity (empirically set to a very small value), $d$ is the L2 distance function and $\hat{P}_{t-1}$ and $P_t^{'}$ are the two image patches at time $t-1$ and $t$ respectively. This equation, in simpler terms, assigns a value of $0$ to $\Psi$, when the two image patches are very similar, making the term $(d(\hat{P}_{t-1}, P_t^{'}) - \xi)$ negative and assigns a positive value when they are not quite similar (thereby indicating presence of motion). The estimated value of $\Psi$ is then mapped to a $0$ or $1$ output, as described in the following equation:

$$\Psi \leftarrow {}^{\Psi}/_{|d(\hat{P}_{t-1}, P_t^{'}) - \xi|} \tag{II}$$

where, $|.|$ denotes the absolute value function.

Finally, $\Psi$ is multiplied with the calculated normalized cross-correlation between two image patches to reduce the possibility of getting a high score in spite of absence of significant motion, thereby making the algorithm more robust. We show results obtained using using this version of the NCCL in figures D.3 and D.4.

## B Experiments with 3D Convolutions

Volumetric (or 3D) convolutions work on spatio-temporal data and can be seen as an extension to the original spatial (or 2D) convolutional networks. As performing convolutions on volumes introduce

---

**Algorithm A.1:** Smoothed normalized cross-correlation score for finding similarity between a set of predicted frame(s) and a set of ground-truth frame(s).

---

**Input**: Ground-truth frames ($GT$), Predicted frames ($PRED$), Gaussian filter ($GLPF$)
      [Dimension = $(h+4) \times (h+4)$]
**Output**: Smoothed Cross-correlation score ($Score_{SNCC}$)
`// ` $h$ `= height and width of an image patch`
`// ` $H$ `= height and width of the predicted frames`
`// ` $t$ `= current time`
`// ` $T$ `= Number of frames predicted`

1  **Initialize:** $Score_{SNCC} = 0$;
2  **for** $t = 1$ *to* $T$ **do**
3     **for** $i = 0$ *to* $H$, $i \leftarrow i + h$ **do**
4         **for** $j = 0$ *to* $H$, $j \leftarrow j + h$ **do**
5             $P_t \leftarrow extract\_patch(PRED_t, i, j, h)$;
             `/* Extracts a patch from the predicted frame at time ` $t$ ` of dimension` $h \times h$ ` starting from the top-left pixel index ` $(i, j)$       `*/`
6             $\hat{P}_{t-1} \leftarrow extract\_patch(GT_{t-1}, i-2, j-2, h+4)$;
             `/* Extracts a patch from the padded ground-truth frame at time` $(t-1)$ ` of dimension ` $(h+4) \times (h+4)$ ` starting from the top-left pixel index ` $(i-2, j-2)$    `*/`
7             $P_t^{'} \leftarrow extract\_patch(PRED_{t-1}, i-2, j-2, h+4)$;
             `/* Extracts a patch from the padded predicted frame at time ` $t$ ` of dimension ` $(h+4) \times (h+4)$ ` starting from the top-left pixel index` $(i-2, j-2)$    `*/`
8             $\Psi \leftarrow max\big(0, d(\hat{P}_{t-1}, P_t^{'}) - \xi\big)$;   `// Checks similarity between two patches`
9             $\Psi \leftarrow \Psi / |d(\hat{P}_{t-1}, P_t^{'}) - \xi|$ ;                  `// ` $\Psi = 0/1$ ;
10            $\hat{P}_{t-1} \leftarrow (GLPF * \hat{P}_{t-1})$ ;        `// Convolve with the Gaussian filter`
11            $\mu_{P_t} \leftarrow avg(P_t)$;
12            $\mu_{\hat{P}_{t-1}} \leftarrow avg(\hat{P}_{t-1})$;
13            $\sigma_{P_t} \leftarrow standard\_deviation(P_t)$;
14            $\sigma_{\hat{P}_{t-1}} \leftarrow standard\_deviation(\hat{P}_{t-1})$;
15            $Score_{SNCC} \leftarrow Score_{SNCC} + \Psi \times max\big(0, \sum_{x,y} \frac{(P_t(x,y)-\mu_{P_t})(\hat{P}_{t-1}(x,y)-\mu_{\hat{P}_{t-1}})}{\sigma_{P_t} \sigma_{\hat{P}_{t-1}}}\big)$;
16         **end**
17     **end**
18     $Score_{SNCC} \leftarrow Score_{SNCC} / \lfloor H/h \rfloor^2$ ;         `// Average over all the patches`
19  **end**
20  $Score_{SNCC} \leftarrow Score_{SNCC} / (T-1)$;            `// Average over all the frames`

---

significant computational complexity, and the multi-stage generative adversarial architecture (refer to sec. 2 in the paper) comprises of two stages, each consisting of two networks, we modified the architecture for incorporating 3D modules. We evaluated these smaller networks with (and without) the proposed objective functions. The network architecture used for this task is shown in table I. Max-pooling is used to scale down the dimensions in the discriminator network to make the training time feasible.

As can be seen from tables IV and V, adding volumetric modules to capture temporal relationship in the data does not produce expected results when coupled with the L1 loss alone. Inclusion of the proposed objective functions *viz.*, NCCL, SNCCL, PCDL and 3-PCDL improves the result by a significant margin. It should be noted that, although the values increase, they remain quite similar to the results obtained from the model discussed in the paper.

Comparing these results to that in table 2 of the paper, it can be inferred that, the performance gain obtained in the quality of predictions, do not generally depend on the type of convolutions used

Table I: Network architecture details. $G$ and $D$ represents the generator and discriminator networks respectively. $U$ denotes a volumetric unpooling operation which upsamples an input by a factor of 2. $M$ represents a max-pooling layer which downsamples an input of by a factor of 2.

| Network | Stage-1 (G) | Stage-2 (G) | Stage-1 (D) | Stage-2 (D) |
|---|---|---|---|---|
| Number of feature maps | 64, 128U, 64 | 64, 128U, 256U, 128, 64 | 64, 128M, 256 | 128, 256M, 128 |
| Kernel sizes | 5, 3, 5 | 5, 5, 5, 5, 5 | 3, 5, 5 | 5, 5, 5 |
| Fully connected | N/A | N/A | 1024, 512 | 1024, 512 |

Table II: Experimental results of applying 3D convolutions in the proposed GAN framework on the UCF dataset. Best results in bold.

| Methods | Frame-1 | | Frame-2 | | Frame-4 | |
|---|---|---|---|---|---|---|
| | PSNR | SSIM | PSNR | SSIM | PSNR | SSIM |
| Adv + L1 | 29.7 | 0.87 | 25.9 | 0.82 | 21.2 | 0.58 |
| Adv + NCCL + L1 | 35.8 | 0.94 | 34.1 | 0.92 | 28.8 | 0.75 |
| Adv + NCCL + PCDL | 37.4 | **0.95** | 35.2 | 0.92 | 29.5 | 0.75 |
| Adv + NCCL + PCDL + L1 | 37.6 | **0.95** | 35.9 | 0.92 | 30.2 | 0.76 |
| Adv + SNCCL + PCDL + 3-PCDL | **38.3** | **0.95** | **36.7** | **0.93** | **31.2** | **0.77** |

and the proposed objective functions directly bridges this gap by introducing the notion of temporal coherence between motion features of the frames.

Table III: Experimental results of applying 3D convolutions in the proposed GAN framework on the KITTI dataset. Best results in bold.

| Methods | Frame-1 | | Frame-2 | | Frame-4 | |
|---|---|---|---|---|---|---|
| | PSNR | SSIM | PSNR | SSIM | PSNR | SSIM |
| Adv + L1 | 30.2 | 0.88 | 26.4 | 0.82 | 22.1 | 0.59 |
| Adv + NCCL | 37.4 | 0.91 | 35.5 | 0.90 | 27.7 | 0.75 |
| Adv + NCCL + PCDL | 39.2 | 0.92 | 36.7 | 0.90 | 29.9 | 0.76 |
| Adv + NCCL + PCDL + L1 | 40.1 | 0.93 | 37.4 | **0.91** | 30.5 | 0.76 |
| Adv + SNCCL + PCDL + 3-PCDL | **40.4** | **0.94** | **37.9** | **0.91** | **31.2** | **0.77** |

## C   Long Distance Prediction

We experimented with the proposed mult-stage GAN model to predict frames far away in the future from the current input frame. For this, we fed the model a sequence of 8 input frames to generate an output sequence of 8 predicted frames. Unlike [A], which uses smaller networks to tackle this situation, we keep our model identical to the one described in section C as the number of parameters to learn in the training stage remains inside feasibility limit. Evaluation results are shown in tables IV-V.

(i) GT (t=8)&emsp;(ii) L1 (t=8)&emsp;(iii) SNCCL (t=8)&emsp;(iv) Combined (t=8)

(v) GT (t=9)&emsp;(vi) L1 (t=9)&emsp;(vii) SNCCL (t=9)&emsp;(viii) Combined (t=9)

(ix) GT (t=10)&emsp;(x) L1 (t=10)&emsp;(xi) SNCCL (t=10)&emsp;(xii) Combined (t=10)

(xiii) GT (t=11)&emsp;(xiv) L1 (t=11)&emsp;(xv) SNCCL (t=11)&emsp;(xvi) Combined (t=11)

(xvii) GT (t=12)&emsp;(xviii) L1 (t=12)&emsp;(xix) SNCCL (t=12)&emsp;(xx) Combined (t=12)

(xxi) GT (t=13)&emsp;(xxii) L1 (t=13)&emsp;(xxiii) SNCCL (t=13)&emsp;(xxiv) Combined (t=13)

(xxv) GT (t=14)&emsp;(xxvi) L1 (t=14)&emsp;(xxvii) SNCCL (t=14)&emsp;(xxviii)Combined (t=14)

(xxix) GT (t=15)&emsp;(xxx) L1 (t=15)&emsp;(xxxi) SNCCL (t=15)&emsp;(xxxii) Combined (t=15)

Figure C.1: Prediction results shown on a clip from the UCF-101 dataset. The model predicts 8 frames given 8 input frames. Best viewed in color.

|  |  |  |  |
|---|---|---|---|
| (i) GT (t=8) | (ii) L1 (t=8) | (iii) SNCCL (t=8) | (iv) Combined (t=8) |
| (v) GT (t=9) | (vi) L1 (t=9) | (vii) SNCCL (t=9) | (viii) Combined (t=9) |
| (ix) GT (t=10) | (x) L1 (t=10) | (xi) SNCCL (t=10) | (xii) Combined (t=10) |
| (xiii) GT (t=11) | (xiv) L1 (t=11) | (xv) SNCCL (t=11) | (xvi) Combined (t=11) |
| (xvii) GT (t=12) | (xviii) L1 (t=12) | (xix) SNCCL (t=12) | (xx) Combined (t=12) |
| (xxi) GT (t=13) | (xxii) L1 (t=13) | (xxiii) SNCCL (t=13) | (xxiv) Combined (t=13) |
| (xxv) GT (t=14) | (xxvi) L1 (t=14) | (xxvii) SNCCL (t=14) | (xxviii)Combined (t=14) |
| (xxix) GT (t=15) | (xxx) L1 (t=15) | (xxxi) SNCCL (t=15) | (xxxii) Combined (t=15) |

Figure C.2: Zoomed in prediction results shown on the clip shown in fig. C.1 from the UCF-101 dataset. The model predicts $8$ frames given $8$ input frames. Best viewed in color.

Table IV: Experimental results of predicting 8 frames given 8 input frames by the proposed GAN framework on the UCF-101 dataset. First four rows are from [A]. Best results in bold.

| Methods | Frame-1 | | Frame-2 | | Frame-4 | |
|---|---|---|---|---|---|---|
| | PSNR | SSIM | PSNR | SSIM | PSNR | SSIM |
| L2 | 18.3 | 0.59 | - | - | 15.4 | 0.51 |
| Adv | 21.1 | 0.61 | - | - | 17.1 | 0.52 |
| L1 | 21.3 | 0.66 | - | - | 17.0 | 0.55 |
| GDL + L1 | 21.4 | 0.69 | - | - | 17.7 | 0.58 |
| Last input | 30.6 | 0.90 | - | - | 21.0 | 0.74 |
| Adv + NCCL + L1 | 35.4 | 0.94 | 33.9 | 0.92 | 22.4 | 0.69 |
| Adv + NCCL + PCDL + L1 | 37.3 | **0.95** | 35.7 | 0.92 | 23.6 | **0.69** |
| **Adv + SNCCL + PCDL + 3-PCDL + L1** | **38.2** | **0.95** | **36.8** | **0.93** | **24.2** | **0.70** |

Table V: Experimental results of predicting 8 frames given 8 input frames by the proposed GAN framework on the KITTI dataset. Best results in bold.

| Methods | Frame-1 | | Frame-2 | | Frame-4 | |
|---|---|---|---|---|---|---|
| | PSNR | SSIM | PSNR | SSIM | PSNR | SSIM |
| Adv + L1 | 29.8 | 0.87 | 26.0 | 0.82 | 20.2 | 0.58 |
| Adv + NCCL | 37.6 | 0.91 | 35.4 | 0.90 | 21.3 | 0.71 |
| Adv + NCCL + PCDL | 39.2 | 0.92 | 36.8 | 0.90 | 21.9 | 0.71 |
| Adv + NCCL + PCDL + L1 | 40.2 | 0.93 | 37.6 | **0.91** | 22.3 | 0.71 |
| Adv + SNCCL + PCDL + 3-PCDL | **40.5** | **0.94** | **38.2** | **0.91** | **22.8** | **0.72** |

# D Prediction Outputs

|  |  |  |  |
| --- | --- | --- | --- |
| (i) Input (t=0) | (ii) Input (t=1) | (iii) Input (t=2) | (iv) Input (t=3) |
| (v) GT (t=4) | (vi) GT (t=5) | (vii) GT (t=6) | (viii) GT (t=7) |
| (ix) L1 (t=4) | (x) L1 (t=5) | (xi) L1 (t=6) | (xii) L1 (t=7) |
| (xiii) SNCCL (t=4) | (xiv) SNCCL (t=5) | (xv) SNCCL (t=6) | (xvi) SNCCL (t=7) |
| (xvii) Combined (t=4) | (xviii) Combined (t=5) | (xix) Combined (t=6) | (xx) Combined (t=7) |
| (xxi) Zoomed GT (t=4) | (xxii) Zoomed GT (t=5) | (xxiii) Zoomed GT (t=6) | (xxiv) Zoomed GT (t=7) |
| (xxv) Zoomed L1 (t=4) | (xxvi) Zoomed L1 (t=5) | (xxvii) Zoomed L1 (t=6) | (xxviii) Zoomed L1 (t=7) |
| (xxix) Zoomed SNCCL (t=4) | (xxx) Zoomed SNCCL (t=5) | (xxxi) Zoomed SNCCL (t=6) | (xxxii) Zoomed SNCCL (t=7) |
| (xxxiii) Zoomed Combined (t=4) | (xxxiv) Zoomed Combined (t=5) | (xxxv) Zoomed Combined (t=6) | (xxxvi) Zoomed Combined (t=7) |

Figure D.3: Prediction results shown on a clip from the $Basketball\_Dunk$ class of the UCF-101 dataset. The model predicts 4 frames given 4 input frames. Figures xxi - xxxvi shows the zoomed in version of an area containing significant motion. $GT$ denotes the ground truth frames at a particular time. $Combined$ represents the loss function described in eqn. 10 of the paper. Best viewed in color.

| | | | |
|---|---|---|---|
| (i) Input (t=0) | (ii) Input (t=1) | (iii) Input (t=2) | (iv) Input (t=3) |
| (v) GT (t=4) | (vi) GT (t=5) | (vii) GT (t=6) | (viii) GT (t=7) |
| (ix) L1 (t=4) | (x) L1 (t=5) | (xi) L1 (t=6) | (xii) L1 (t=7) |
| (xiii) SNCCL (t=4) | (xiv) SNCCL (t=5) | (xv) SNCCL (t=6) | (xvi) SNCCL (t=7) |
| (xvii) Combined (t=4) | (xviii) Combined (t=5) | (xix) Combined (t=6) | (xx) Combined (t=7) |
| (xxi) Zoomed GT (t=4) | (xxii) Zoomed GT (t=5) | (xxiii) Zoomed GT (t=6) | (xxiv) Zoomed GT (t=7) |
| (xxv) Zoomed L1 (t=4) | (xxvi) Zoomed L1 (t=5) | (xxvii) Zoomed L1 (t=6) | (xxviii) Zoomed L1 (t=7) |
| (xxix) Zoomed SNCCL (t=4) | (xxx) Zoomed SNCCL (t=5) | (xxxi) Zoomed SNCCL (t=6) | (xxxii) Zoomed SNCCL (t=7) |
| (xxxiii) Zoomed Combined (t=4) | (xxxiv) Zoomed Combined (t=5) | (xxxv) Zoomed Combined (t=6) | (xxxvi) Zoomed Combined (t=7) |

Figure D.4: Prediction results shown on a clip from the $Javelin\_Throw$ class of the UCF-101 dataset. The model predicts $4$ frames given $4$ input frames. Figures xxi - xxxvi shows the zoomed in version of an area containing significant motion. $GT$ denotes the ground truth frames at a particular time. $Combined$ represents the loss function described in eqn. 10 of the paper. Best viewed in color.

(i) Input (t=0)    (ii) Input (t=1)    (iii) Input (t=2)    (iv) Input (t=3)

(v) GT (t=4)    (vi) GT (t=5)    (vii) GT (t=6)    (viii) GT (t=7)

(ix) L1 (t=4)    (x) L1 (t=5)    (xi) L1 (t=6)    (xii) L1 (t=7)

(xiii) SNCCL (t=4)    (xiv) SNCCL (t=5)    (xv) SNCCL (t=6)    (xvi) SNCCL (t=7)

(xvii) Combined (t=4)    (xviii) Combined (t=5)    (xix) Combined (t=6)    (xx) Combined (t=7)

(xxi) Zoomed GT (t=4)    (xxii) Zoomed GT (t=5)    (xxiii) Zoomed GT (t=6)    (xxiv) Zoomed GT (t=7)

(xxv) Zoomed L1 (t=4)    (xxvi) Zoomed L1 (t=5)    (xxvii) Zoomed L1 (t=6)    (xxviii) Zoomed L1 (t=7)

(xxix) Zoomed SNCCL (t=4)    (xxx) Zoomed SNCCL (t=5)    (xxxi) Zoomed SNCCL (t=6)    (xxxii) Zoomed SNCCL (t=7)

(xxxiii) Zoomed Combined (t=4)    (xxxiv) Zoomed Combined (t=5)    (xxxv) Zoomed Combined (t=6)    (xxxvi) Zoomed Combined (t=7)

Figure D.5: Prediction results shown on a clip from the KITTI dataset. The model predicts 4 frames given 4 input frames. Figures xxi - xxxvi shows the zoomed in version of an area containing significant motion. *GT* denotes the ground truth frames at a particular time. *Combined* represents the loss function described in eqn. 10 of the paper. Best viewed in color.

# References

[A] M. Mathieu, C. Couprie, and Y. LeCun. Deep multi-scale video prediction beyond mean square error. *International Conference on Learning Representations (ICLR)*, 2016.