[Reviews · NeurIPS 2017]

Reviewer 1



This method provides 2 contributions for next frame prediction from video sequences. The first is the introduction of a normalized cross correlation loss, which provide a better similarity score to judge if the predicted frame is close to the true future. The second is the pairwise contrastive divergence loss, based on the idea of similarity of the image features. Results are presented on the UCF101 and Kitti datasets, and a numerical comparison using image similarity metrics (PSNR, SSIM) with Mathieu et al ICLR16 is performed. Comments: The newly proposed losses are interesting, but I suspect a problem in the evaluation. I fear that the evaluation protocol comparing the present approach to previous work [13] is different to [13]: In the table, the authors don't mention evaluating their method on 10% of the test set and only in the moving area as Mathieu et al. did. Did the authors performed the sanity check of comparing the obtained PSNR/SSIM score of frame copy to see if they match the ones reported in [13]? Please be more specific about the details of the evaluation (full test set? Full images? -> in this case, what are the scores for a frame copy baseline?) The authors answered my concerns in the rebuttal. - Providing generated videos for such paper submission would be appreciated. In the displayed examples, including the ones from the supplementary material, the movements are barely visible. - Predicting several frames at a time versus one? In Mathieu et al. work, results showed that predicting only one frame at the time worked better. Have you tried such experiment? l.138: I don't understand the h+4? Minor The first sentence, "Video frame prediction has always been one of the fundamental problems in computer vision as it caters to a wide range of applications [...]" is somehow controversial in my opinion: The task was, in the best of my knowledge, introduced only in 2014. As for the applications, even if i believe they will exist soon, I am not aware of systems using next frames predictions yet. l. 61 uses -> use l. 65 statc -> statistical ? l 85. [5]is -> [5] is Eq 4: use subscripts instead of exponents for indexes l 119 (b)P -> (b) P l 123-125 weird sentence l. 142 ,u -> , u l. 153 tries to -> is trained to l. 163 that, -> that l. 164 where, -> where [13] ICLR 16 [17] ICLR 16 please check other references

Reviewer 2



The paper presents a method for predicting the future frames of video sequences. It contributes with a multi-stage GAN approach, including new loss functions to the GAN community specialized to the video domain. These loss functions enforce local and global smoothness over time. The paper has several positive and negative sides. Positive sides: 1. It is addressing an interesting and relevant problem, that is going to be of wide interest, especially in the computer vision community. 2. It is relatively novel. The new things are comming through the usage of the contrastive loss and the cross-correlation loss as well as the multi-stage GAN idea. 3. Relatively well written and easy to read. 4. Positive experimental evaluation. The results on the UCF dataset suggest that the combined method is better than the baselines, and the different components do contribute to a better overall performance. Negative sides: 1. Shallow experimental section. While the paper has an extensive experimental evaluation in the supplementary material, the experiments section in the paper is quite short and shallow. The paper could leave out some of the details from the previous sections and dedicate more space for the experiments. The paper says that it conducts experiments on three datasets, however, only one is reported in the paper. 2. Questionable experimental conduct. The reported results in the paper are on UCF, but the model is trained on Sports1M. Why the cross-dataset transfer? Is that comparable to the method of [13]? The paper should make sure the setup is the same as for the previous works. If cross-dataset transfer is needed, then how about seeing results on other pairs of datasets? 3. State-of-the-art comparison. The paper compares to state-of-the-art on the UCF dataset and to [13] only. It would be beneficial to see compatisons to other works on other datasets too. 3. Analysis. The paper would greatly benefit from a deeper analysis of different segments of the method. What are also the typical bad, failure cases? What is the maximum reasonable temporal extent at test time? How deep in the future can the method generalize when it is trained on X frames?

Reviewer 3



This paper proposes a coarse-to-fine pipeline with two-stage GANs for video frames prediction. For each stage, the generator takes as input a sequence of image patches of low-resolution and outputs a sequence of higher resolution patches. To further alleviate issues of unstable training of GAN and blurry frames, authors also compute the sum of discriminator's decisions on all frames and add two terms in the loss to encourage temporal coherence. Quantitative and qualitative evaluation on real-world video datasets demonstrate effectiveness of the proposed method. 1. In Figure 1, output of stage 2 has size of 128x128 but input size is only 32x32. According to section 2.1, height and width of output should be 2 times of input, therefore it should be 64x64 instead of 128x128. 2. As for NCC loss, I wonder why not compute that for the current predicted frame and the current ground-truth frame instead of the previous one. Besides, how is the square patch size h determined? How to handle the case when the motion of a patch is beyond the small local neighborhood? How much time does it take to compute NCC loss for a minibatch? 3. In section 6.2, authors mentioned that random sequences of 32x32 patches are extracted from frames with significant motion for evaluation, but they did not explain how many sequences of patches are extracted from those frames for each video. In addition, there is very little analysis about the experimental result, especially the qualitative result. 4. It is better to add some intermediate visualization results to get more insight of the proposed two loss terms. For example, authors can visualize NCC similarity of a sequence and get a a heatmap from which the influence of NCC similarity can be checked. Author can also visualize p_i, p_{i+1} and corresponding Y^{\hat}_i and y^{\hat}_{i+1}.